New interpretation of the cranial osteology of the Early Cretaceous turtle Arundelemys dardeni (Paracryptodira) based on a CT-based re-evaluation of the holotype

Evers Serjoscha W. serjoscha.evers@googlemail.com serjoscha.evers@unifr.ch
Rollot Yann
Joyce Walter G.
Department of Geosciences, University of Fribourg , Fribourg , Switzerland
Young Mark
Electronic publication date: 2021 May 31
Publication date: 2021
Volume: 9
Electronic Location ID: e11495
Received 2021 Mar 24; Accepted 2021 Apr 30
Copyright: ©2021 Evers et al.
Copyright year: 2021
Copyright holder: Evers et al.
License: This is an open access article distributed under the terms of the Creative Commons Attribution License, which permits unrestricted use, distribution, reproduction and adaptation in any medium and for any purpose provided that it is properly attributed. For attribution, the original author(s), title, publication source (PeerJ) and either DOI or URL of the article must be cited.
License URL: https://creativecommons.org/licenses/by/4.0/

Keywords: Testudines, Paracryptodira, Baenidae, Cranial scutes, Turtles, CT-scan, Turtle evolution, Cranial anatomy

Funding: Swiss National Science Foundation SNF 200021_178780/1 The work is supported by a grant from the Swiss National Science Foundation (SNF 200021_178780/1). The funders had no role in study design, data collection and analysis, decision to publish, or preparation of the manuscript.

==============================
Arundelemys dardeni is an Early Cretaceous paracryptodire known from a single, incomplete, but generally well-preserved skull. Phylogenetic hypotheses of paracryptodires often find Arundelemys dardeni as an early branching baenid. As such, it has a central role in understanding the early evolution of the successful clade Baenidae, which survived the Cretaceous–Paleogene mass extinction, as well as the diversification of Paracryptodira into its subclades, which recent research suggests to perhaps include helochelydrids, compsemydids, pleurosternids, and baenids. Computer tomography scans of the holotype material that were produced for the initial description of Arundelemeys dardeni reveal several errors in the initial anatomical description of the species, which we correct based on element-by-element segmentation. In addition, we provide entirely novel anatomical information, including descriptions of several previously undescribed cranial bones, the endosseous labyrinth, and the cranial scutes, the latter of which are unknown for most paracryptodires. We provide an interpretation of cranial scutes which homologizes the scutes of Arundelemys dardeni with those of other stem turtles.

Introduction

Ardundelemys dardeni was originally described on the basis of a single, partial cranium by Lipka et al. (2006). These authors also described the cranial anatomy with reference to computed tomography (CT) scans, albeit without providing element-by-element segmentations of the specimen (USNM 497740). As was common at the time, the descriptions in Lipka et al. (2006) were kept relatively brief and some cranial elements (e.g., prootic, opisthotic, exoccipital) were not described at all, despite being preserved. Based on a number of traits, such as the positioning of the foramen for the internal carotid artery, Lipka et al. (2006) identified Arundelemys dardeni as an early branching paracryptodire, but most phylogenetic analyses have since retrieved this species as an early baenid (Lyson et al., 2011; Lyson et al., 2016; Lyson & Joyce, 2011; Larson et al., 2013; Lively, 2015; Pérez-García et al., 2015; Smith et al., 2017; Joyce & Rollot, 2020; Rollot, Evers & Joyce, in press). As part of an on-going project to better understand paracryptodire anatomy, systematics, and evolution, we recently (re-)described the cranial anatomy of Eubaena cephalica (Rollot, Lyson & Joyce, 2018), Pleurosternon bullockii (Evers, Rollot & Joyce, 2020) and Uluops uluops (Rollot, Evers & Joyce, in press). During this work, we found a number of discrepancies between the published anatomy of Arundelemys dardeni with our own perceptions of the anatomy of the specimen, which led us to produce a full cranial segmentation of the holotype material on the basis of the CT scan already used in the initial study. Here, we present the resulting cranial models, and provide a re-description of the cranial anatomy of Arundelemys dardeni that corrects some errors of the previous work of Lipka et al. (2006), which, for the most part, remains an accurate account of the gross morphology of the material. Although our observations provide novel anatomical information and detail that could inform paracryptodire phylogeny, we only provide a description of Arundelemys dardeni without further phylogenetic work, as we are currently preparing a comprehensive phylogenetic analysis for the group.

Material & Methods

We used the high-resolution X-ray computed tomography (CT) scans of USNM 497740 produced by Lipka et al. (2006) for our segmentations. These data are available on the online repository Digimorph, where Matthew Colbert is indicated as the person who obtained the scans at the University of Texas in Austin in 2004. The specimen was scanned at a beam energy of 180 kV, a current of 0.133 mA, and without using a filter. The voxel size data for the scan are non-isotropic, with x and y pixels being 0.04346 mm, and the interslice spacing in the z-plane (coronal plane) being 0.09457 mm. The resulting CT-scans were segmented in the software Mimics (v., 19.0; http://biomedical.materialise.com/mimics). Besides segmenting all individual bones, we segmented the left labyrinth, as well as the paths of the carotid arteries and selected cranial nerves of USNM 497740. 3D models were exported as .ply files. Figures of digital renderings were compiled using the software Blender v. 2.71 (blender.org). CT-slice data as well as the 3D models are deposited at MorphoSource (Evers, 2021). The comparative 3D models of Uluops uluops are also digitally available at MorphoBank (Rollot, Evers & Joyce, in press; DOI 10.7934/P3919).

Systematic Palaeontology

TESTUDINATA Klein, 1760	
PARACRYPTODIRA Gaffney, 1975	
ARUNDELEMYS Lipka et al., 2006	
Arundelemys dardeniLipka et al., 2006	

Holotype: USNM 497740, a nearly complete cranium.

Type locality and horizon: USNM locality 41614 (Hotton locality), an open-pit clay mine near Muirkirk, Maryland, USA; Potomac Formation, Early Cretaceous, late Albian–early Aptian. See Lipka et al. (2006) for additional geological information.

Figure 1 Three dimensional renderings of the cranium of Arundelemys dardeni (USNM 497740).

(A) Dorsal view. (B) Ventral view. (C) Left lateral view. (D) Right lateral view. (E) Anterior view. (F) Posterior view. Abbreviations: boc, basioccipital; cap, carotid pit; epi, epipterygoid; ex, exoccipital, f, frontal; fpp, foramen palatinum posterius; fprp, foramen prepalatinum; fst, foramen stapedio-temporale; ica, incisura columellae auris; j, jugal; lir, lingual ridge; mdaf, mandibular artery foramen; mx, maxilla; n, nasal; op, opisthotic; pa, parietal; pal, palatine; pbs, parabasisphenoid; pm, premaxilla; po, postorbital; pr, prootic; prf, prefrontal; pt, pterygoid; pte, processus pterygoideus externus; q, quadrate; soc, supraoccipital; tf, trigeminal foramen; v, vomer. Note that bone labels are in bold. Scale bars equal 10 mm.

Revised diagnosis: Arundelemys dardeni can be diagnosed as a member of Paracryptodira based on the presence of a characteristic combination of derived and symplesiomorphic features. These include the presence of skull sculpturing (also present in some sinemydids; Brinkman & Peng, 1993; Zhou, Rabi & Joyce, 2014; and some xinjiangchelyids: Tong et al., 2019); a relatively strong lateral orbit orientation; the retention of relatively large nasals paired with their partial posterior separation by the anterior frontal processes (the nasals of early Testudinata or Meiolaniformes are in medial contact with one another across their entire length: Gaffney, 1990; Sterli, 2015; and the nasals of sinemydids are small: e.g., Rabi et al., 2013; Li et al., 2019); a unique combination of jugal features that includes a dorsally raised jugal position (as in xinjiangchelyids, but unlike in meiolaniforms; varies in sinemydids: see Brinkman & Wu, 1999; Li et al., 2019) and its exclusion from the orbital margin (unlike in sinemydids, xinjiangchelyids, meiolaniforms); retention of a posteriorly open incisura columellae auris; and a unique combination of traits surrounding the embedding of the carotid artery, including the presence of a carotid pit, absence of internal carotid artery embedding, and the absence of a palatine artery canal or interpterygoid vacuity indicating the reduction of the respective artery (retained in Uluops uluops among paracryptodires: Rollot, Evers & Joyce, 2021). A distinct lingual ridge that is better developed along the anterior half of the palate, a ventral process of the jugal that nearly contributes to the labial margins, and an expanded posterior process of the pterygoid that contacts the basioccipital and exoccipital are characteristics typical of baenids, but the potential baenid affinities of Arundelemys dardeni should be tested in a phylogenetic analysis.

Description

General comments. Although differences in interpretation between our study and that of Lipka et al. (2006) are pointed out individually in the respective description sections below, the reader shall note that our biggest re-interpretations of morphology concern (i) the preservation of the temporal roof and supraoccipital, which we interpret to be far less complete; (ii) the morphology of the anterior end of the pterygoids, which we interpret as forming extended anterior processes rather than forming a straight transverse suture with the anterior part of the palate (Fig. 1); (iii) the shape, position and contacts of the epipterygoid (Fig. 1); and (iv) the placement of the foramina anterius canalis carotici basisphenoidalis.

Figure 2 Cranial scutes of Arundelemys dardeni (USNM 497740).

(A) Three dimensional rendering of the cranium in dorsal view. (B) Interpretative line drawing of A. (C) Three dimensional rendering of the cranium in left lateral view. (D)Interpretative line drawing of C. Note that sutural lines are thin black lines in B & D, whereas thick lines are scute sulci, which are labelled with capital letters (following the nomenclature of Sterli & Fuente, 2013). Abbreviations: epi, epipterygoid; f, frontal; j, jugal; mx, maxilla; n, nasal; op, opisthotic; pa, parietal; pm, premaxilla; po, postorbital; pr, prootic; prf, prefrontal; pt, pterygoid; q, quadrate; soc, supraoccipital. Scale bar equals 10 mm.

The skull surface of USNM 497740 is weakly sculptured by low and irregular pits (Lipka et al., 2006), which are present across all skull roof and temporal bone elements as well as the maxilla. This texture is similar to that in Trinitichely hiatii (SW Evers ,Y Rollot, WG Joyce, 2021, pers. obs.), but distinct from the dense tubercular pattern in Uluops uluops (Rollot, Evers & Joyce, in press), the fine crenulations in Dorsetochelys typocardium (DORCM G 23), or the striated pattern of Pleurosternon bullockii (Evers, Rollot & Joyce, 2020). Overall, the sculpturing in Arundelemys dardeni is less uniform, less dense, and less strongly structured than in other early paracryptodires.

Figure 3 Proposed homology of cranial scutes for selected stem turtles.

(A) Proganochelys quenstedtii in dorsal view (based on Gaffney, 1990; Sterli & Fuente, 2013). (B) Mongolochelys efremovi in dorsal (top) and lateral (bottom) view (based on Khosatzky, 1997; Sukhanov, 2000; Sterli & Fuente, 2013). (C) Meiolania platyceps in dorsal (top) and lateral (bottom) views (modified from Gaffney 1983; Sterli & Fuente, 2013). Note that D and H scutes are switched with regard to previous figures of Meiolania. (D) Arundelemys dardeni in lateral (top) and dorsal (bottom) view. Note that skull was mirrored for comparison. (E) Ordosemys leios in dorsal view (based on Li et al., 2019). Note additional identification of H scute. (F) Annemys levensis in dorsal view (modified from Rabi et al., 2014). Note changes in D and H scute position and midline contact of anterior F scute. Capital letter labels indicate scute names. Skulls are not to scale. Note that fossil preservation did not allow identification of scutes for lateral skull surfaces of turtles shown in A, E–F.

In addition to surface sculpturing, several scute sulci can be traced across the skull roof of USNM 497740 (Fig. 2). These were mentioned by Lipka et al. (2006), but not described. The sulci are relatively broad but low grooves spread across the dermal skull roof, but only few sulci can be followed for their full extent. In the 3D models, they can best be appreciated with low light/at low angles and when rotating the specimen. For ease of communication about individual scutes, we applied the scute nomenclatural system of Sterli & Fuente (2013). This system was developed as a homology hypothesis of scutes, and has primarily been applied to compare the scute patterns of early shelled turtles, including Proganochelys quenstedtii, with meiolaniforms (Sterli & Fuente, 2013; Sterli, 2015), but has since been used for turtles of other clades as well (e.g., Rabi et al., 2013; Rabi et al., 2014; Li et al., 2019). In this context, it is important to note that the concept presented in Sterli & Fuente (2013: their fig. 10) seems to have an inconsistency regarding the identification of scutes H and D. Specially, scutes H and D are switched in panel C (Meiolania platyceps) with regard to all other turtles shown. In particular, whereas the D scute is anteroventrally in contact with the F scute series and covers the parietal-postorbital region in all other turtles (including other derived meiolaniids), the respective scute in Meiolania platyceps is labelled as ‘H’ (Sterli & Fuente, 2013: their fig. 10). The scute posterolaterally to this position is the H scute in all other turtles, but labelled as ‘D’ in Meiolania platyceps. The pattern of Meiolania platyceps used by Sterli & Fuente (2013); is identical to that used Gaffney (1983), which itself is based on Simpson (1938). Thus, the scute identification for Meiolania platyceps has historical roots, and it is unfortunate that the Meiolania-pattern has apparently been incorrectly applied to the other turtles shown by Sterli & Fuente (2013). If this Meiolania-based nomenclature were given priority, all other taxa shown by Sterli & Fuente (2013; their fig.10) as well as all following publications which identified cranial scutes in turtles (e.g., Rabi et al., 2013; Rabi et al., 2014; Li et al., 2019) would be in consecutive error. We find it easier to correct this discrepancy by changing the pattern for Meiolania platyceps than by revising all descriptions that followed the incorrect translation of D vs. H scute identification. Thus, we establish the following comparative criteria to identify the D and H scutes in turtles: the D scute is in contact with the F-scute series, is positioned anterolaterally to scute H, and commonly extends across parts of the parietal-postorbital suture. The H scute does not contact the F-scute series, is positioned posteromedially to scute D, contacts the posterior midline scutes (A, and usually but not universally X), and does not extend anteriorly onto the postorbital. These criteria allow consistent D vs. H scute identification among the taxa shown by Sterli & Fuente (2013; their fig.10) and following authors (Fig. 3). In the future, the H and D scutes of previous Meiolania platyceps illustrations (Gaffney 1983: their fig. 23; Sterli & Fuente, 2013: their fig.10; (Sterli, 2015): their fig. 1) should be read in reverse, as shown here in Fig. 3. Besides the aforementioned change for Meiolania platyceps, our scute identification for Annemys levensis (Fig. 3) deviates slightly from the original one presented by Rabi et al. (2014): their fig.1K, and we add the identification of an H scute to Ordosemys leios (Li et al., 2019).

Because not all scute sulci show clearly on USNM 497740, the following reconstruction for Arundelemys dardeni should be seen as tentative (Figs. 2 and 3). None the less, the scute pattern and scute homology across clades is something that may be rewarding in terms of future phylogenetic assessments, and we hope that the following instigates more in-depth research on turtle cranial scutes.

The skull of Arundelemys dardeni is covered, for the most part, by relatively large scutes, which is more similar to the pattern seen in derived meiolaniids (e.g., Gaffney, 1983; Sterli, 2015) than the pattern of early testudinatans (e.g., Gaffney, 1990; Sterli & Joyce, 2007). A single, large scute covered the anterior part of the skull anterior to the level of the orbits (Figs. 2B, 2D), which corresponds to scute Z in the homology concept of Sterli & Fuente (2013). Immediately posterior, there seems to be another large unpaired scute (scute Y) that covers the anterior frontal and prefrontal region (Figs. 2B, 2D). The skull midline posterior to scute Y seems to covered by three consecutive scutes. There are no sagittal median sulci that would suggest that any of these midline scutes were paired, but the absence of such sulci cannot be considered strong evidence for their absence, given the discontinuous scute sulcus pattern in the specimen. The first of these three scutes, tentatively identified as scute G, is restricted to the frontals, and, if indeed unpaired, much broader than long (Fig. 2B). In other turtles for which the scute patterns have been reported, scute G is usually paired across the midline (Gaffney, 1983; Gaffney, 1990; Sterli & Fuente, 2013). The posteriorly following scute in USNM 497740 (scute X) is somewhat larger than scute G, but still broader than long, covering the posterior aspect of the frontals and most of the preserved parietal region (Fig. 2B). This scute usually indeed is unpaired, but it is usually restricted to the parietals (i.e., it does not extend anteriorly onto the frontals) and is often smaller than reconstructed here for Arundelemys dardeni (Sterli & Fuente, 2013). The last preserved median scute of USNM 497740 (scute A), is restricted to the posterior interparietal region (Fig. 2B), and posteriorly incomplete due to breakage in the skull roof. Its anterior, preserved morphology corresponds well with that reported for Mongolochelys efremovi (Khosatzky, 1997; Sukhanov, 2000).

The scutes in the temporal region around the postorbital are relatively well defined on the left side of USNM 497740. The posterior part of the prefrontals and the lateral frontal process that extends into the orbit is covered by a relatively small scute (scute F1; Figs. 2B, 2D). The left and right scutes F1 are separated by scute Y. Posterior to scute F1, there is a pentagonal scute that covers the posterolateral frontal process and the anterior postorbital region, and extends into the orbital margin. We identify this scute as a second F scute (F2), as circumorbital scutes in this region of the orbit are limited to scute F in the concept of Sterli & Fuente (2013), which in non-meiolaniid taxa is generally developed as a series of scutes. Topological arguments with respect to anteriorly and laterally adjacent scutes support this identification: the respective scute of USNM 497740 has (i) an anterior contact with scute F1; (ii) an anteromedial contact with scute G; and (iii) a lateral contact with scute D. Our interpretation results in the observation that the F2 scute of Arundelemys dardeni has an additional, medial contact with scute X, which also prevents a contact between scutes D and D. This is not seen in other turtles for which the scute patterns have been reported (Sterli & Fuente, 2013; Rabi et al., 2013; Rabi et al., 2014; Sterli, 2015; Li et al., 2019). However, this can be explained by the large size of the X scute in Arundelemys dardeni, which can be identified with high certainty, and which is much larger than in other turtles (Sterli & Fuente, 2013; Rabi et al., 2014). Additional, tentative evidence for our identification comes from the baenid Neurankylus torrejonensis (Lyson et al., 2016). Although scute identities are tricky for this taxon, as there seems to be a partial reduction of midline scutes, the scute that covers the postorbital parietal region posterior to the circumorbital scutes can be identifies as a D scute, and it also lacks a contact with the anterior midline scutes. Thus, the absence of a D–G scute contact may be common feature of paracryptodires. Posterior to our scute F2, there is a moderately sized scute in Arundelemys dardeni that covers the parietal and postorbital suture, which we identify as scute D (Figs. 2B, 2D), following the arguments laid out for D scute identification above. Between scutes D and A, there is a scute that as preserved is restricted to the parietals of USNM 497740 (Fig. 2B), and which can be tentatively identified as scute H, again following above arguments.

Figure 4 Three dimensional renderings of the nasal cavity and orbitotemporal region of Arundelemys dardeni (USNM 497740).

(A) Medial view on left half of anterior cranium, showing the inside of the nasal cavity. (B) Posterolateral view of cranium, viewing into the temporal fossa. Abbreviations: boc, basioccipital; crcr, crista cranii; dpp, descending process of parietal; epi, epipterygoid; ex, exoccipital; f, frontal; fja, foramen jugulare anterius; ica, incisura columellae auris; ina, internal naris; j, jugal; jmp, jugal medial process; lar, labial ridge; lir, lingual ridge; mdaf, mandibular artery foramen; mr, medial ridge of prefrontal; mx, maxilla; n, nasal; op, opisthotic; pa, parietal; pm, premaxilla; po, postorbital; pppt, posterior process of pterygoid; prf, prefrontal; pro, prootic; pte, processus pterygoideus externus; q, quadrate; soc, supraoccipital; sot, septum orbitotemporale; tf, trigeminal foramen; vr, ventral ridge of nasal. Scale bar equals 10 mm.

The lateral surface of the skull also shows scute sulci. The orbito-temporal region is dorsally covered by a large scute tentatively identified as scute F3 based on its dorsal contact with scutes F2 and D, its contact with the orbital margin, and its position that is restricted to the postorbital (see Sterli & Fuente, 2013). Ventral to scute F3, USNM 497740 has two scutes between the orbit and cheek emargination (Fig. 2D). Both are herein attributed to the J-series. J1 is a small scute in the orbital margin, which covers the postorbital-maxilla contact (Fig. 2D). Posteriorly adjacent is the larger J2, which completely covers the jugal and its sutures with the postorbital and maxilla (Fig. 2D). Along the maxilla, no sulci can be inferred except those defining the J-series as well as the one defining scute Z ventrally. Thus, it seems that the maxilla and premaxilla were covered by a single large scute I (Fig. 2D).

Nasal. Both nasals of USNM 497740 are preserved (Figs. 1A, 1F). The nasal of Arundelemys dardeni has a relatively large dorsal exposure. It contacts the frontal posteriorly, the prefrontal posterolaterally, and the maxilla laterally. Medially, there is a short inter-nasal contact that is posteriorly prohibited by the anterior processes of the frontals (Fig. 1A). The nasal is horizontally aligned with the plane of the frontal, and does not slope anteroventrally like in Uluops uluops Rollot, Evers & Joyce, 2021 or Pleurosternon bullockii (Evans & Kemp, 1975; Evers, Rollot & Joyce, 2020). Instead, the anterior margin of the nasal, which forms the dorsal margin of the external naris, is anteriorly oriented, contributing to a high snout region similar to that of Trinitichelys hiatti (Gaffney, 1972). On their ventral surface, there is a broad, transverse ridge crossing both nasals (Fig. 4A), which is absent in Uluops uluops.

Prefrontal. Both prefrontals are preserved in USNM 497740 (Figs. 1A, 1C–1D). The prefrontal of Arundelemys dardeni contacts the frontal medially, the nasal anteriorly, the maxilla anterolaterally, the vomer ventromedially, and the palatine posteroventromedially. As described by Lipka et al. (2006), the prefrontal has a rectangular exposure on the skull roof (Fig. 1A). However, this prefrontal exposure is reduced with regard to most turtles, similar to the condition of Uluops uluops (Rollot, Evers & Joyce, 2021) and Pleurosternon bullockii (Evers, Rollot & Joyce, 2020), but not as strongly as in baenodds (e.g., Gaffney, 1972; Lyson & Joyce, 2009a; Lyson & Joyce, 2009b; Rollot, Lyson & Joyce, 2018; Lyson, Sayler & Joyce, 2019). Left and right elements are separated by broad anterior frontal processes. The ventral process of the prefrontal is transversely broad and frames a narrow fissura ethmoidalis. The process has a clear contact with the palatine, which could not be ascertained by Lipka et al. (2006). The prefrontal forms the dorsal border of a larger foramen orbito-nasale, which is otherwise framed by the maxilla and the palatine. An unusual feature of the prefrontal of Arundelemys dardeni is a medially projecting sheet-like ridge on the medial surface that faces the inside of the nasal cavity (Fig. 4A). This ridge is absent in Uluops uluops (Rollot, Evers & Joyce, 2021) or Pleurosternon bullockii (Evers, Rollot & Joyce, 2020).

Frontal. Both frontals are well preserved in USNM 497740 (Figs. 1A, 1C–1D, 1F). The frontal of Arundelemys dardeni is distinct in its shape. The anterior process of the frontal, which extends medial to the prefrontal and contacts the nasal anteriorly, is very broad, similar to the condition in Uluops uluops (Rollot, Evers & Joyce, 2021), but different from the particularly narrow processes seen in Pleurosternon bullockii (Evans & Kemp, 1975; Evers, Rollot & Joyce, 2020). The process is parallel with its lateral margin to the sagittal plane for the length of the prefrontal, but anteriorly tapers toward the midline, forming with its counterpart a distinct “V”-shaped projection that separates the nasals (Fig. 1A). The anterior frontal process extends dorsally to roof the nasal cavity and extends the sulcus olfactorius into the nasal cavity (Fig. 4A). The cristae cranii on the ventral surface, which define the sulcus olfactorius are distinct ridges (Fig. 4A), but become shallow posteriorly so that they are not confluent with the anteroventral margin of the descending process of the parietal. The frontal has a short lateral process that contributes to the orbital margin between the prefrontal and postorbital (Figs. 1A, 1C–1D), as noted by Lipka et al. (2006). Posterolaterally, the frontal extends between the parietal and postorbital, forming a broad pointed process (Fig. 1A). Consequently, the suture with the parietal is not approximately mediolaterally oriented, as in Pleurosternon bullockii (Evers, Rollot & Joyce, 2020) and Uluops uluops (Rollot, Evers & Joyce, 2021), but instead crosses the skull roof obliquely from the contact with the postorbital anteromedially, as in baenodds such as Baena arenosa, Chisternon undatum, Eubaena cephalica, Palatobaena cohen, or Saxochelys gilberti (Gaffney, 1972; Lyson & Joyce, 2009a; Rollot, Lyson & Joyce, 2018; Lyson, Sayler & Joyce, 2019).

Parietal. Both parietals are incompletely preserved in USNM 497740 (Figs. 1A, 1C–1F). This assessment is in contrast to what was reported by Lipka et al. (2006), who interpreted the parietals to be unbroken in the parietal-postorbital region and extremely deep dorsal skull emarginations to be present. Lipka et al. (2006) cite two primary reasons for their interpretation, in particular the absence of trabecular intersections along the margins of the purported upper temporal emargination and the thinning of the bones towards the margins. In our reassessment of CT scans, we find numerous trabecular intersections along the parietal margins, contra Lipka et al. (2006). In addition, the thinning of the parietal toward its margin observed by Lipka et al. (2006) is irregular and not symmetrical for both skull sides (Figs. 1E, 4B). These observations, therefore, indicate that the preserved parietal margin is the result of breakage. Comparative anatomical reasons support the hypothesis that the temporal roof is damaged. In all paracryptodires from which this region is known, the parietals form an expanded posteromedian process that overlaps the supraoccipital almost completely, even in taxa with moderately deep emarginations (e.g., Pleurosternon bullockii: Evans & Kemp, 1975; Evers, Rollot & Joyce, 2020) or deep emarginations (e.g., Plesiobaena antiqua: Brinkman, 2003, Palatobaena cohen: Lyson & Joyce, 2009a). As preserved, this process is absent in Arundelemys dardeni. Additionally, in all known paracryptodires, including palatobaenins with comparatively deep emarginations (Joyce & Lyson, 2015), the posterior and posterolateral parietal margin prohibits most of the prootics to be seen in dorsal view. Arundelemys dardeni would be the only paracryptodire, in which the prootics are completely exposed in dorsal view (Fig. 1A). Finally, significant damage is apparent to the supraoccipital and quadrates as well (see below). Thus, the combined observations presented here suggest that both the postorbitals, parietals, and jugals are missing their posterior portions, making it impossible to assess the depth of dorsal emarginations in Arundelemys dardeni.

The preserved parts of the parietal show contacts with the frontal anteriorly, the postorbital anterolaterally, the other parietal across the skull midline, the supraoccipital posteroventrally, the prootic posteroventrolaterally, the epipterygoid ventrally along the descending process, and a possible but short contact with the crista pterygoidei of the pterygoid medially to the epipterygoid (Figs. 1A, 1C–1F). The skull roof as formed by the parietals slopes upwards posteriorly. Ventrally, the parietal forms a descending process, which overlaps the anterolateral surface of the supraoccipital posteriorly, articulates tightly with the prootic, forms the dorsal margin of the trigeminal foramen posterior to its contact with the epipterygoid, and forms the anterior margin of braincase, which also functions as the posterior margin of the foramen interorbitale (Figs. 1C–1D). It is unclear if the parietal would have excluded the prootic from the trigeminal foramen via a posteroventral process, as in in Pleurosternon moncayensis (Pérez-García et al., 2021). As recently reported for Pleurosternon bullockii (Evers, Rollot & Joyce, 2020), the lateral surface of the descending parietal process bears a strong lateral ridge (Fig. 4B), which contacts the postorbital and defines the dorsal aspects of a septum orbito-temporale with it, similar to the condition seen in pleurodires. Thus, orbital and temporal fossae are clearly delimited in Arundelemys dardeni.

Postorbital. Both postorbitals are incompletely preserved in USNM 497740 (Fig. 1) and the reader is referred to the parietal section above for arguments as to why we think that significant portions of the postorbital are broken off posteriorly. What remains of the postorbital laterally contacts the frontal and parietal in the skull roof (Fig. 1A). A quadratojugal contact is not preserved but was likely present, but a possible contact with the squamosal cannot be assessed. Ventrally, the postorbital contacts the maxilla, jugal, and pterygoid along a medially expanded ventral process (Figs. 1C–1D). The medial surface of this process is formed as a septum orbitotemporale, a strong ridge separating the orbital and temporal cavities (Fig. 4B). This is similar to pleurodires and Pleurosternon bullockii (Evers, Rollot & Joyce, 2020), but in Arundelemys dardeni, this wall of bone is even more extensive than in Pleurosternon bullockii. Though not described under that name, a well-developed septum orbitotemporale is also apparent among baenodds such as Eubaena cephalica (Rollot, Lyson & Joyce, 2018). Ventromedially, the septum orbitotemporale extends all the way to the transverse process of the pterygoid. This is only preserved on the right side of the specimen, but very clearly visible in the 3D models. The ventral postorbital process also forms the majority of the posterior margin of the orbit (Figs. 1C–1D), where it has a direct contact with the maxilla, excluding the jugal from the orbital margin.

Jugal. Both jugals are preserved in USNM 497740 (Figs. 1C–1D). The posterior margin of both jugals appears partly broken, so that a possible contact with the quadratojugal cannot be assessed. However, a short piece of the posteroventral margin of the jugal seems to be intact on either side of the skull. This part of the lateral plate of the jugal nearly reaches the labial margin of the maxilla. Instead of being part of the labial margin sensu stricto, the jugal in this region forms a posterodorsally sloping continuation of the labial margin (Figs. 1C–1D). This sloping edge usually forms the anterior margin of the cheek emargination in all paracryptodires but compsemydids (Lyson & Joyce, 2011), and the sloping edge may thus be seen as tentative evidence that the cheek emargination was similarly moderately deep as in baenids (Joyce & Lyson, 2015) and pleurosternids (Evers, Rollot & Joyce, 2020). Although the jugal of Arundelemys dardeni is thus positioned somewhat dorsally to the ventral skull margin as formed by the labial margin of the maxilla, the ventral extend of the jugal is deeper than in Pleurosternon bullockii (Evers, Rollot & Joyce, 2020), Uluops uluops (Rollot, Evers & Joyce, 2021), and instead more similar to the condition seen in Palatobaena cohen (Lyson & Joyce, 2009a) or Plesiobaena antiqua (Brinkman, 2003), although this is not the universal condition in baenids (e.g., Trinitichelys hiatti: Gaffney, 1972; Eubaena cephalica: Rollot, Lyson & Joyce, 2018). As already observed by Lipka et al. (2006), the jugal is very clearly excluded from contributing to the orbital margin by a postorbital maxilla contact (Figs. 1B–1C). Nonetheless, the jugal is broadly exposed in the posterior floor of the orbital fossa (Fig. 4A). Here, it forms the ventral extension of the septum orbitotemporale (Fig. 4B). Like the postorbital, the jugal has a clear medial contact with the pterygoid (Fig. 4B). In fact, the jugal posteromedially even extends slightly onto the transverse process of the pterygoid, or more specifically, its horizontally exposed lateral flange.

Quadratojugal. Both quadratojugals are not preserved in USNM 497740 and the quadrates are too incompletely preserved to infer their presence based on respective articulation facets. Given that other paracryptodires have quadratojugals, their absence in Arundelemys dardeni most likely is a preservational artefact.

Squamosal. Both squamosals are not preserved in USNM 497740.

Premaxilla. Both premaxillae are completely preserved in USNM 497740 (Figs. 1B–1D, 1F). The premaxilla forms the anteroventral tip of the snout, and contacts the maxilla laterally, and the vomer posteriorly, and its counterpart medially. Several features of the premaxilla of Arundelemys dardeni are noteworthy. The bone is anteroposteriorly elongate, contributing to an extensive ventral exposure (Fig. 1B). The premaxilla is relatively broad and completely surrounds the foramen praepalatinum (Fig. 1B), a feature also seen in Pleurosternon bullockii (Evers, Rollot & Joyce, 2020) and some baenodds (Gaffney, 1972). This condition differs strongly from the premaxilla described for Compsemys victa (Lyson & Joyce, 2011), which is transversely narrow in ventral view. USNM 41614 also differs in the triturating surface ridges from Compsemys victa: the labial ridge of USNM 497740 is low and even and continuous with the maxilla, as the vast majority of paracryptodires. A ventrally recurved hook, as developed in Compsemys victa (Lyson & Joyce, 2011), is absent. A lingual ridge is very clearly developed. It is discontinuous at the median interpremaxillar contact (Fig. 1B), but becomes deeper posterolaterally toward the maxilla, where the ridge becomes extremely prominent and deeper than the labial ridge (Fig. 1F). The lingual ridges define a median depression in the triturating surface formed by the premaxilla, maxilla, and vomer, that is commonly referred to as the “tongue groove” and found in numerous baenodds, including Baena arenosa (Gaffney, 1972), Eubaena cephalica (Rollot, Lyson & Joyce, 2018), “Plesiobaena” antiqua (Brinkman, 2003).

Maxilla. Both maxillae of USNM 497740 are well preserved (Figs. 1B–1D, 1F). The bone has a posterior process that extends ventral to the orbit to contact the postorbital, jugal, and pterygoid. Along its medial margin, the maxilla broadly contacts the palatine, and, anterior to the foramen orbito-nasale, the vomer and the premaxilla (Fig. 1B). Along its ascending process, which frames the orbit anteriorly, the maxilla contacts the nasal, and, posterior to this contact, the prefrontal (Figs. 1C–1D).

The medial surface of the ascending process of the maxilla is dominated by a medially projected ridge, which abuts the prefrontal. Ventrally, the ridge very clearly defines the nasal ducts that connect the nasal cavity with the internal naris (Fig. 4A). The ridge slopes ventrally lower within the orbital cavity, where it forms the floor of the latter, contributes to the foramen orbito-nasale, and extends posteriorly to the pterygoid.

As in many paracryptodires, the ventral margin of the maxilla forms a slightly convex labial ridge (Figs. 1C–1D). The labial ridge is deep posteriorly, but becomes slightly shallower anteriorly. Over the entire extent of the maxilla, the triturating surface is deeply grooved between the labial ridge laterally and the lingual ridge medially (Figs. 1B, 4A). Lipka et al. (2006) noted relatively large openings in the triturating surface of the maxilla, which they interpreted, after also considering breakage, as “architectural features reflecting the distribution of stresses through the triturating surface” (Lipka et al., 2006: p. 304). We here interpret these openings as a result of erosion (Fig. 1B). As already noted by Lipka et al. (2006), the CT scan shows that the maxillary bone is thin in this region, and this seems to be safely attributable to abrasion. Additionally, the openings are only superficially bilaterally symmetrical (Fig. 1B), arguing against being part of the natural morphology of Arundelemys dardeni. As a final argument, it is biomechanically hard to conceive how holes in the surfaces for food processing should mitigate stresses from processing food on these very surfaces: finite element analysis of turtle skulls which simulate biting behaviour show that the triturating surfaces between labial and lingual ridges are the least stressed skull snout regions during biting, and that high stresses are actually concentrated along palatal openings, including the foramen prepalatinum, foramen palatinum posterius, or internal naris (Ferreira et al., 2020). The lingual ridge of USNM 497740 is very strongly developed (Lipka et al., 2006; Fig. 1B, 1F, 2A), which is unlike the condition in Pleurosternon bullockii (Evans & Kemp, 1975; Evers, Rollot & Joyce, 2020) or Uluops uluops (Rollot, Evers & Joyce, 2021). The lingual ridge of USNM 497740 becomes deeper anteriorly and is most prominent at the level of the articulation with the vomer. The better development of the lingual ridge in the anterior half of the palate is a feature commonly found among baenodds, including Baena arenosa (Gaffney, 1972), Plesiobaena antiqua (Brinkman, 2003), or Stygiochelys hiatti (Gaffney, 1972). Medial to the labial ridge, the maxilla slopes dorsally toward the vomer in this region, which results in a morphology in which the vomer-premaxilla contact seems deeply embedded between the lingual ridges in ventral view, the aforementioned “tongue groove” (see Premaxilla above) typical of baenodds (Fig. 1B).

Posteromedially, the maxilla contacts the palatine (Fig. 1B). Posterior to this contact, the maxilla has a short contribution to the foramen palatinum posterius, before the maxilla contacts the anterolateral portion of the transverse process of the pterygoid.

Vomer. The vomer is a singular, median bone with only minor damage in USNM 497740 (Fig. 1B). The vomer forms a broad anteroventral surface between the articulation with the right and left maxillae to form a “tongue groove” (see Premaxilla above). This surface anteriorly contacts the premaxillae, with which it forms a short triturating surface between the maxillary lingual ridges (Fig. 1B). Posteriorly, the vomer extends from this surface as a posteriorly narrowing process that is wedged between the palatines and that contacts the pterygoids with its posterior tip (Fig. 1B). The ventral surface of the vomer is without a medial keel. A sulcus vomeri is formed on the dorsal surface of the vomer between the anterolateral processes that contact the prefrontals.

Palatine. Only the right palatine is preserved in USNM 497740 (Fig. 1B). The palatine is a flat element that is centrally constricted between its anteroposteriorly broadened medial and lateral margins. The lateral margin contacts the maxilla and minorly contributes to the lingual ridge (Lipka et al., 2006), which posteriorly becomes broad and low (Fig. 1B). Medially, the palatine lies against the vomer anteriorly and the pterygoid posteriorly, so that a median interpalatine contact is absent (Lipka et al., 2006). A short anterior contact with the prefrontal is also present. The foramen palatinum posterius is anteriorly framed by the palatine (Fig. 1B) and the concave anterior margin of the bone defines the posterior margin of the internal naris.

Quadrate. Both quadrates of USNM 497740 are incompletely preserved (Figs. 1B 1A–1E). In addition, the sutures with the prootic and opisthotic are very tight and only distinct in places in the CT scan, resulting in some sutural approximation in our segmentations, particularly with the opisthotic (note stippled sutural lines in Fig. 2). Besides contacts with these two bones, the quadrate preserves a contact with the pterygoid. Contacts with the quadratojugal and squamosal were near certainly present, but this cannot be confirmed due to the absence of these bones or the appropriate sutures.

The preserved parts of the quadrate of USNM 497740, particularly on the left side, show that the incisura columellae auris was a widely open posteriorly (Figs. 1C, 4B), which is distinct from the narrow slits of Dorsetochelys typocardium (Evans & Kemp, 1976), Pleurosternon bullockii (Evans & Kemp, 1975; Evers, Rollot & Joyce, 2020), Pleurosternon moncayensis (Pérez-García et al., 2021), Uluops uluops (Rollot, Evers & Joyce, 2021) or baenids (Gaffney, 1972). It is also distinct from the morphology of Compsemys victa, in which the incisura becomes posterolaterally enclosed by a contact between a posterodorsal process of the quadrate and a posteroventral process of the squamosal (Lyson & Joyce, 2011). Although the latter is not preserved in USNM 497740, the posterior surface of the left quadrate is well-preserved and lacks features that would indicate that a posterodorsal quadrate process is broken or that the squamosal would have extended far ventrally to contact the quadrate. The stapedial canal is formed between the quadrate and prootic on both sides, but its dorsal aperture, the foramen stapedio-temporale, has an additional contribution of the opisthotic on the left side (absent on right side), showing polymorphism in this feature (Fig. 1B). The anterior quadrate contact with the pterygoid is clear in the CT scans and they show than an anteriorly directed epipterygoid process is absent in Arundelemys dardeni (Fig. 1C).

Epipterygoid. Both epipterygoids are well preserved in USNM 497740 (Figs. 1C–1D, 4B) and clearly visible in the CT slices. However, our interpretation of the epipterygoid position and shape differs strongly form that of Lipka et al. (2006). These authors figure the epipterygoid as a rod-like element (their Fig. 2H), which forms the ventral margin of the trigeminal foramen with its dorsal margin and extends posteroventrally to the quadrate from there. However, this configuration would be highly unusual, as the epipterygoid of turtles usually ossifies between the crista pterygoidei of the pterygoid and the descending process of the parietal anterior to the trigeminal foramen, and not posterior to it (Gaffney, 1979a; Gaffney, 1979b). In addition, an epipterygoid-quadrate contact is only present in turtles when an anteriorly directed, thin epipterygoid process of the quadrate is present (Gaffney, 1979a; Gaffney, 1979b), which is not the case in Arundelemys dardeni. The CT scans of USNM 497740 instead show a more usual epipterygoid shape and position for Arundelemys dardeni: the epipterygoid is ossified between the descending process of the parietal and the crista pterygoidei of the pterygoid, prohibiting a broad contact between these bones (Figs. 1C–1D, 4B). It forms the anteroventral margin of the trigeminal foramen and contacts the pterygoid posteriorly, but not the quadrate. An unusual feature of the epipterygoid of Arundelemys dardeni is a large opening along its ventral contact with the pterygoid (Figs. 1C–1D, 4B). This opening is symmetrically present on both sides of the skull and the CT scans do not indicate any damage in this region, from which we gather that the opening may represent a true foramen. The foramen forms a connection between the sulcus cavernosus and the temporal fossa. We are only aware of a similar opening in the Cretaceous sandownid turtle Sandownia harrisi (Evers & Joyce, 2020), in which the respective foramen was interpreted as a potential foramen for the mandibular artery, which, in many turtles, extends through the canalis cavernosus alongside the lateral head vein until it exits either through a separate foramen posterior to the trigeminal foramen, or through the trigeminal foramen itself (see Rollot, Evers & Joyce, 2021).

Pterygoid. The left pterygoid of USNM 497740 is completely preserved (Fig. 1B), whereas the right element is abraded posteriorly. As in all turtles, the pterygoid morphology is complex with many processes contributing to different structures. Anteriorly, the pterygoid of Arundelemys dardeni has a conspicuous anteromedial process, which medially contacts the other pterygoid forming a long anterior spur that contacts the vomer and which separates the palatines along the skull midline (Figs. 1B, 5A). Such processes are present in other early paracryptodires, for example Uluops uluops (Rollot, Evers & Joyce, 2021). Posterior to the level of the processes, but still along the anterior part of the bone, the pterygoid of USNM 497740 has a laterally directed transverse process, which contacts the maxilla, jugal and postorbital (Figs. 1B, 4B). The lateral surface of the transverse process is expanded to the typical vertical flange, but the tip of the process, in general, is much smaller than in Uluops uluops (Rollot, Evers & Joyce, 2021) or Pleurosternon bullockii (Evers, Rollot & Joyce, 2020). However, similar to these taxa, the vertical flange of the transverse process is slightly medially recurved along its dorsal margin, establishing a slight reminiscence of the pleurodire condition of this process.

Figure 5 Three dimensional renderings of the basicranial region of Arundelemys dardeni (USNM 497740).

(A) pterygoids and parabasisphenoid in ventral view. (B) as A, but bones rendered transparent and carotid artery and facial nerve models added. (C) anterodorsal view of pterygoids, parabasisphenoid, carotid artery, facial nerve, and abducens nerve. Abbreviations: app, anterior process of pterygoid; cap, carotid pit; cera, cerebral artery; faccb, foramen anterius canalis carotici basisphenoidalis; faf, fossa acustico-facialis; fdnv, foramen distalis nervi vidiani; facnv, foraman anterius canalis nervi vidiani; fpcnv, foramen posterius canalis nervi vidiani; gg, geniculate ganglion; ica, internal carotid artery; pbs, parabasisphenoid; pt, pterygoid; pte, processus pterygoideus externus; rbp, retractor bulbi pit; scav, sulcus cavernosus; tf, trigeminal foramen; VI, abducens nerve; VII, facial nerve; VIIhyo, hyomandibular branch of the facial nerve; VIIvi, vidian branch of the facial nerve; VIII, acoustic nerve. Scale bar equals 10 mm.

The interpterygoid contact of USNM 497740 is long (Lipka et al. 2016), but right and left pterygoids diverge mid-length to make room for the median parabasisphenoid, and, more posteriorly, the basioccipital (Figs. 1B, 4B). The posterior process of the pterygoid is very extensive in Arundelemys dardeni, extending over the full length of parabasisphenoid and basioccipital, also contacting the exoccipital (Fig. 1E). This is different from the condition in Uluops uluops, in which the pterygoid only extends to the level of the basioccipital (Rollot, Evers & Joyce, 2021), but similar to the condition of baenodds (e.g., Rollot, Lyson & Joyce, 2018). The posterior pterygoid process of USNM 497740 fully covers the cavum acustico-jugulare in ventral view, and has a notably deep pterygoid fossa between quadrate and parabasisphenoid (Lipka et al., 2006; Fig. 1B). Along the suture with the parabasisphenoid, the pterygoid contributes to a carotid pit (see parabasisphenoid; Fig. 5). Just posterior to the carotid pit, there is a small foramen distalis nervi vidiani and associated canalis pro ramo nervi vidiani (sensu Rollot, Lyson & Joyce, 2018; Figs. 5A–5B) for the posterior course of the vidian nerve. This canal traverses the pterygoid dorsoventrally, and connects the canalis cavernosus with the ventral skull surface. In Uluops uluops (Rollot, Evers & Joyce, 2021) and Pleurosternon bullockii (Evers, Rollot & Joyce, 2020) there is a small foramen posterius canalis nervi vidiani (sensu Rollot, Lyson & Joyce, 2018) in the anterior region of the carotid pit through which the vidian nerve enters the pterygoid to extend through a long canal traversing the bone. In USNM 41616, such a foramen is also apparent within the carotid pit (Fig. 5B), albeit hard to see in the CT scans. The associated canalis nervus vidianus is also hard to trace through the CT scans, particularly on the right side, but we were able to segment the canal for nearly its full length on both sides. The anterior opening foramen (foramen anterius canalis nervi vidiani) is located on the dorsal pterygoid surface anterior to the base of the crista pterygoidei and close to the position of the foramen palatinum posterius (Fig. 5C).

The dorsal surface of the posterior pterygoid process is exposed in the floor of the cavum acustico-jugulare, where it forms a broad anteroposteriorly directed groove for the course of the lateral head vein. Together with the prootic, the pterygoid forms the canalis cavernosus for the anterior course of the lateral head vein. At the level of the anterior end of the canalis cavernosus, the pterygoid has a dorsally raised crista pterygoidei, which forms the ventral margin of the trigeminal foramen (Figs. 1C–1D, 4B, 5C). Anterior to the trigeminal foramen, the pterygoid broadly contacts the epipterygoid, and forms the ventral border of the possible mandibular artery foramen (see epipterygoid) that opens in the lateral wall of the braincase between both bones (Figs. 1C–1D, 4B). Medial to the crista pterygoidei, the dorsal pterygoid surface bears a distinct sulcus cavernosus (Fig. 5C). The anterior abducens nerve foramina are positioned in the suture between pterygoid and parabasisphenoid (Fig. 5C), as is also the case in Pleurosternon bullockii (not stated in Evers, Rollot & Joyce, 2020, but apparent from their models). In Uluops uluops, these foramina even lie entirely within the pterygoid (Rollot, Evers & Joyce, 2021). This lateral placement of the anterior abducens nerve foramina in the aforementioned taxa is highly unusual, as these foramina are usually positioned in the parabasisphenoid in all other known turtles (Gaffney, 1979a; Gaffney, 1979b; Rollot, Evers & Joyce, 2021). Their position has not yet been clarified in baenodds.

Supraoccipital. The supraoccipital is incompletely preserved in USNM 497740 (Figs. 1A, 1C–1E), missing nearly its entire crest. This differs from the interpretation of Lipka et al. (2006), whose descriptions suggest the supraoccipital were basically complete. For example, Lipka et al. (2006) state that Arundelemys dardeni has a short but sharp-edged supraoccipital crest that slopes from the parietals to the foramen magnum. However, this entire dorsal edge is broken (Figs. 1C–1D) and a broad triangular break above the foramen magnum indicates that a relatively large piece of the bone is missing in this area. As such, it is impossible to assess the length of the supraoccipital crest. As already discussed in the parietal section, the dorsal edge of the supraoccipital as preserved would have been overlain by posterior parietal processes, as in other paracryptodires, preventing the current complete dorsal exposure of the bone (Fig. 1A).

The ventral part of the supraoccipital of USNM 497740 is expanded over the braincase to form its roof, and contacts, from posterior to anterior, the exoccipital, opisthotic, prootic, and parietal (Fig. 1A, 1E).

Exoccipital. Both exoccipitals of USNM 497740 are preserved (Fig. 1E), but the morphology of this bone was not described in the original description (Lipka et al., 2006). The exoccipital forms the lateral margin of the foramen magnum, but right and left elements neither meet in the dorsal, nor in its ventral margin (Fig. 1E). As in Uluops uluops (Rollot, Evers & Joyce, in press) or Pleurosternon moncayensis (Pérez-García et al., 2021), the exoccipitals do not seem to contribute to the occipital condyle –although the condyle is abraded in the specimen, the posterior processes of the exoccipitals end shortly posterior to the level of the foramen magnum, but this could be the result of damage as well. The ventral footplate of the exoccipital that abuts the basioccipital is relatively high and forms a nearly circular foramen jugulare anterius with the opisthotic (Fig. 4B). The exoccipitals of Arundelemys dardeni have a broadly developed contact with the pterygoid.

Basioccipital. The basioccipital of USNM 497740 is preserved with the exception of the occipital condyle, which appears to be missing (Fig. 1B). It contacts the parabasisphenoid anteriorly, the pterygoid laterally, the exoccipital laterodorsally, and has a small contact with the opisthotic along the area of the hiatus acusticus between the braincase and cavum labyrinthicum. The basioccipital is much broader mediolaterally than it is long anteroposteriorly (Fig. 1B). The ventral surface is gently excavated by a shallow fossa between the basioccipital tuberculae, which are low mounts in the basioccipital-pterygoid contact area. A very low basis tuberculi basalis is developed on the dorsal surface of the basioccipital at the contact with the anteriorly adjacent parabasisphenoid. We can find no evidence of a canalis basioccipitalis, a canal of unclear function found in many baenodds (Rollot, Lyson & Joyce, 2018).

Prootic. Both prootics are preserved in USNM 497740 (Fig. 1A, 1C–1D), but the prootic morphology was not described in Lipka et al. (2006). As usual in turtles, the prootic contacts the supraoccipital dorsally, the opisthotic posteriorly, the quadrate laterally, the pterygoid ventrally, the parabasisphenoid medioventrally, and the parietal anterodorsally. The anterodorsal surface of the prootic, which is exposed within the temporal fossa, is transversely concavely flexed to form the anterior half of the poorly developed otic trochlea. Similarly indistinct otic trochlear are also present in other early paracryptodires, including Pleurosternon bullockii (Evers, Rollot & Joyce, 2020) and Uluops uluops (Rollot, Evers & Joyce, in press). As preserved, the prootic of Arundelemys dardeni contributes to the trigeminal foramen (Figs. 1C–1D). This differs from the condition of Pleurosternon moncayensis (Pérez-García et al., 2021), in which the prootic is excluded from contributing to the foramen by a posteroventral ramus of the parietal. This process may be broken off in Arundelemys dardeni (see left side, Fig. 1C), but may also be genuinely absent.

Internally, most of the prootic is excavated for the cavities that constitute the anterior part of the cavum labyrinthicum. Ventrally, the prootic is expanded in the floor of the cavum labyrinthicum and forms much of its floor. The fenestra ovalis is completely surrounded by the prootic and opisthotic, i.e., there is a ventral contact with the processus interfenestralis of the opisthotic, as for instance also in Pleurosternon moncayensis (Pérez-García et al., 2021). The pericapsular recess on the posterior surface of the prootic immediately lateral to the fenestra ovalis (see Evers & Benson, 2019) is very well developed in Arundelemys dardeni, and both deeper and broader than in Pleurosternon bullockii (Evers, Rollot & Joyce, 2020) or Uluops uluops (Rollot, Evers & Joyce, 2021). The recess is developed directly dorsal to the posterior entrance foramen of the canalis cavernosus, which is roofed by the prootic. In this dorsal roof, Arundelemys dardeni has a very clearly developed sulcus for the hyomandibular branch of the facial nerve (VII; see Rollot, Evers & Joyce, 2021). Another feature associated with the facial nerve is the canalis nervus facialis, which extends mediolaterally through the prootic from the fossa acustico-facialis into the canalis cavernous (Figs. 5B–5C). This morphology implies that the geniculate ganglion and facial nerve split into vidian and hyomandibular branches was located in the canalis cavernosus (Rollot, Evers & Joyce, 2021), which is also corroborated by the presence of a canalis pro ramo nervi vidiani slightly more anteriorly within the canalis cavernosus and the pterygoid (Fig. 5B). The acoustic nerve has a short canal/foramen from the fossa acustico-facialis directly into the cavum labyrinthicum (Fig. 5C).

Opisthotic. Both opisthotics of USNM 497740 are partially preserved (Figs. 1A, 1E), but were not described initially by Lipka et al. (2006). The left opisthotic is more complete than the right one, and shows all of the contacts with other cranial elements: the supraoccipital dorsomedially, the prootic anteriorly, the quadrate laterally, the exoccipital posteroventrally, and the pterygoid ventrally. Additionally, a short contact with the basioccipital is present anterior to the position of the foramen jugulare anterius. The opisthotic forms the posterior portion of the cavum labyrinthicum. It completely forms the lateral semicircular canal (i.e., the prootic portion of the canal remains medially open toward the cavum), and the processus interfenestralis forms a footplate in the floor of the cavum labyrinthicum. The fenestra ovalis is completely embraced by the prootic and opisthotic, and thus ventrally closed. The fenestra perilymphatica is only incompletely preserved on the left side, but opens as usually from the cavum labyrinthicum into the recessus scalae tympani posteriorly. The opisthotic closes the foramen jugulare anterius anteriorly, with the posterior part being formed by the exoccipital (Fig. 4B). The left opisthotic partially preserves foramina for the glossopharyngeal nerve (IX), both an incomplete lateral foramen at the base of the processus interfenestralis, and a medial foramen opening from the braincase into the cavum labyrinthicum are apparent.

The paroccipital process is relatively short (Fig. 1E), and the sutures toward the quadrate are not entirely clear due to the tight interdigitation of both bones. This results in slightly odd sutural lines between both models (see dotted sutural lines in Fig. 2).

On the right side, the opisthotic is excluded from contributing to the foramen stapedio-temporale, whereas the bone straddles the aperture for the stapedial canal on the left side (Fig. 1A).

Parabasisphenoid. The parabasisphenoid of USNM 497740 has the usual contacts with the basioccipital posteriorly, the prootic dorsolaterally, and the pterygoid lateroventrally (Fig. 1B). The parabasisphenoid of USNM 497740 has the typical shape seen in non-pleurodiran turtles, being anteroposteriorly longer than mediolaterally wide and acutely triangular, tapering anteriorly between the pterygoids (Fig. 1B). The posterior suture with the basioccipital is straight in USNM 497740, and posterior processes that lap onto the basioccipital and form a secondary pair of basioccipital tubera, features seen in Pleurosternon bullockii, Uluops uluops, and helochelydrids (Joyce et al., 2011; Evers, Rollot & Joyce, 2020; Rollot, Evers & Joyce, 2021), are absent. Arundelemys dardeni shares with Pleurosternon bullockii and Uluops uluops the presence of a carotid pit in the lateral contact area with the pterygoid (Figs. 5A–5B). The carotid pit is similar in position and shape to the fenestra caroticus (sensu Rabi et al., 2013) of some turtles, but differs from it in that there is no posterior bony coverage of the internal carotid artery (Fig. 5B). Like in Pleurosternon bullockii (Evers, Rollot & Joyce, 2020) and Uluops uluops (Rollot, Evers & Joyce, 2021) the carotid pit is a broadly recessed area around the entry foramen for the cerebral artery. The respective foramen and canal pierce the parabasisphenoid anteromedially from the carotid pit (Figs. 5A–5B). Following recent advances in understanding and changes in nomenclature of the carotid arterial system of turtles (Rollot, Evers & Joyce, 2021), we describe the foramen not as a foramen posterius canalis carotici interni (as done by Lipka et al., 2006), but as a foramen posterius canalis carotici basisphenoidalis following the ductus of Rollot, Evers & Joyce (2021), who did not define this foramen explicitly but re-named osteological structures for the cerebral artery after the bone it traverses, i.e., the parabasisphenoid. Thus, the foramen posterius canalis carotici basisphenoidalis of Rollot, Evers & Joyce (2021) is equivalent to the foramen posterius canalis carotici cerebralis of Rabi et al. (2013) and the foramen posterius canalis carotici interni of Lipka et al. (2006). Importantly, this particular anatomy suggests that the internal carotid artery of Arundelemys dardeni was completely exposed along the ventral skull surface, that only the cerebral artery enters the canal in the basisphenoid, and that a palatine artery must have either been uncovered in bone, or, more likely, was absent given the absence of a respective canal through the pterygoid-parabasisphenoid region (see pterygoid).

Unlike in Pleurosternon bullockii or Uluops uluops, the parabasisphenoid of USNM 497740 lacks distinct lateral processes at the level of the carotid pit that could be comfortably labelled as remnant basipterygoid processes. Rather than having such processes inserting laterally into the pterygoid and forming the surface for the carotid pit, as in Pleurosternon bullockii (Evers, Rollot & Joyce, 2020) and Uluops uluops (Rollot, Evers & Joyce, 2021), the suture between parabasisphenoid and pterygoid of Arundelemys dardeni is dominated by short to medium-length interdigitating bone spurs.

Dorsally, the parabasisphenoid of USNM 497740 has roughly parallel lateral margins toward the prootics, and the space between them is deeply transversely concave (Fig. 5C). Posteriorly, there is a low basis tuberculi basalis, which is posteriorly confluent with the crista dorsalis basioccipitalis of the basioccipital. Anteriorly, the parabasisphenoid cup that hold the pons of the hindbrain is finished by a low, step-like dorsum sellae that abruptly slopes vertically into the sella turcica (Fig. 5C). This fossa for the pituitary is only shallowly posteriorly excavated, and holds two foramina anterius canalis carotici basisphenoidalis (sensu Rollot, Evers & Joyce, 2021) for the cerebral arteries. Lipka et al. (2006) described the cerebral artery to exit the basisphenoid via a singular foramen, as indeed observed in some turtles (Hooks III, 1998; Evers & Benson, 2019; Evers, Barrett & Benson, 2019; Rollot, Evers & Joyce, 2021b), but we observe very clearly distinctly separate apertures (Fig. 5C). This misinterpretation probably resulted from examining coronal slices anterior to the true position of the foramina, which would lead to seeing a large cross-sectional area that is the posterior part of the sella turcica itself. To either side of the sella turcica, the parabasisphenoid of USNM 497740 has short and relatively stout anteriorly directed clinoid processes (Fig. 5C). Retractor bulbi pits are present underneath the clinoid process base, but small on either side (Fig. 5C). The anterior abducens nerve foramen lies in the lateral margin of the retractor bulbi pit, at the contact with the pterygoid (Fig. 5C). Although not explicitly stated by Evers, Rollot & Joyce (2020), their models of Pleurosternon bullockii show the same morphology, whereas the anterior abducens nerve canal foramina lie even further laterally, completely in the pterygoid in Uluops uluops (Rollot, Evers & Joyce, 2021). This highly unusual feature can probably be used in future phylogenetic or systematic assessments, but its distribution across the tree will need to be established first. Anterior to the sella turcica, the parabasisphenoid of USNM 497740 continues as a flat but relatively broad rostrum basisphenoidale (Fig. 5C), but it terminates a long way before the anterior pterygoid processes, thus not reaching the vomer.

Stapes. The stapes is not preserved in USNM 497740.

Labyrinth. Although the labyrinth of USNM 497740 (Fig. 6) conforms to the general labyrinth morphology of turtles of having a roughly pyramidal outline in lateral view with nearly symmetrical vertical semicircular canals (Georgi & Sipla, 2008; Walsh et al., 2009; Neenan et al., 2017; Lautenschlager, Ferreira & Werneburg, 2018; Evers et al., 2019), there are distinct differences with the labyrinth of Uluops uluops (Rollot, Evers & Joyce, 2021), but similarities to the labyrinth of Pleurosternon moncayensis (Pérez-García et al., 2021). A detailed comparison with Pleurosternon moncayensis is complicated by the sheared preservation of the latter. Overall, the vertical semicircular canals of Arundelemys dardeni are more symmetrical than the ones of Uluops uluops, with the anterior semicircular canal being only slightly longer than the posterior one (Fig. 6A). The vertical semicircular canals are also more rounded along their entire extent. The ‘M’-shaped embayment at the common crus that was described for Pleurosternon moncayensis (Pérez-García et al., 2021) and which is also present in Uluops uluops is inconspicuous in Arundelemys dardeni (Fig. 6A). The ampulla for the lateral semicircular canal is not distinctly apparent in the 3D model of USNM 497740, whereas it shows as a dorsally expanded area in Uluops uluops (Rollot, Evers & Joyce, 2021). In Arundelemys dardeni, all semicircular canals are relatively slender and have small cross-sectional circumferences (Fig. 6), which is as in Pleurosternon moncayensis (Pérez-García et al., 2021), but contrasts with the relatively thick canals in Uluops uluops. A secondary common crus is present in Arundelemys dardeni (Fig. 6B), but the course of the lateral semicircular duct can be approximated in the dorsal region as a weak imprint in the wall of the secondary common crus. The fenestra ovalis shows as a larger, subcircular plane in the 3D model (Fig. 6A), and the fenestra perilymphatica has the usual shape and position in the posterior surface of the lagena area (Fig. 6B). The differences in labyrinth shape between Arundelemys dardeni and Pleurosternon moncayensis on one side and Uluops uluops on the other cannot be easily functionally interpreted for the time being, and undermine the hypothesis of Pérez-García et al. (2021) that the labyrinth shape of pleurosternids is indicative of their freshwater aquatic ecology. It is none the less interesting to note significant shape disparity among relatively closely related species with probable similar ecologies, which highlights that labyrinth ecomorphology is best interpreted in a quantitative comparative framework (Bronzati et al., 2021).

Figure 6 Three dimensional renderings of left labyrinth of Arundelemys dardeni (USNM 497740).

(A) lateral view. (B) posterior view. (C) dorsal view. Abbreviations: asc, anterior semicircular canal; cc, common crus; fov, fenestra ovalis; fpl, fenestra perilymphatica; lsc, lateral semicircular canal; psc, posterior semicircular canal; scc, secondary common crus.

Discussion

Although this re-description of Arundelemys dardeni is based on the same material and CT scans as the original description by Lipka et al. (2006), we re-interpret several features of the anatomy, pertaining to distinct skull regions (skull roof, palate, braincase wall, internal arterial canals). Our deviating interpretations are backed up by digital renderings of the skull from CT data, although some re-interpretations, such as the anterior pterygoid shape, resulted from disagreeing that the interpretative cranial line drawings provided in the original study matched the photographs shown in Lipka et al. (2006) and can thus be validated without the use of CT technology. It is currently unclear how the updated anatomical interpretations provided here affect phylogenetic assessments, but our interpretations certainly change character scorings. Arundelemys dardeni was initially described as a basal paracryptodire, but primarily on the basis of carotid features that received considerable re-interpretation since Lipka et al. (2006). Here, we show that many features that allegedly show close correspondence between Arundelemys dardeni and Compsemys victa (see Lipka et al., 2006), such as the extent of the temporal emargination, were probably the result of misinterpretations. Instead, the more detailed account of the cranial anatomy of Arundelemys dardeni given here shows some intriguing similarities with Late Jurassic pleurosternids (e.g., position of anterior foramina for the abducens nerve, presence of a well-developed carotid pit), although many differences to pleurosternids are also clear, for instance in the absence of secondary basioccipital tubera formed by posterior parabasisphenoid processes. Many similarities to baenids are apparent as well, including the probable early baenid Trinitichelys hiatti. This is for instance the case in the shared presence of a ventral process of the jugal that nearly contacts the labial margin of the maxilla, a distinct lingual ridge that is better developed near the contact of the maxilla with the vomer, a straight basisphenoid-basioccipital suture, extensive contact of the pterygoid with the exoccipital, the clear absence of a canal for the palatine artery (which is present in the pleurosternid Uluops uluops but not Pleurosternon bullockii). Many of the recently observed cranial features of paracryptodires have yet to be integrated into systematic work as phylogenetic characters, and this paper provides additional comparative information to do that. This includes our attempt at documenting and homologizing cranial scutes for paracryptodires. Scute sulci are often not visible in non-baenid paracryptodires due to intense skull sculpturing (e.g., Uluops uluops), but the retained pattern in Arundelemys dardeni might help in elucidating paracryptodiran relationships among global turtle phylogeny, as it represents a thus far unused set of anatomical characters and may provide novel signals.

Conclusions

The cranial anatomy of Arundelemys dardeni is re-interpreted and we offer differing views to those provided in the initial description of the only available skull. Instead of having a largely completely preserved skull roof with extreme emarginations and dorsally completely exposed supraoccipital and prootics, we interpret Arundelemys dardeni as missing significant portions of the dorsal skull roof formed by the parietals, postorbitals, and supraoccipital. These and other changes in anatomical interpretation have no effect on the paracryptodiran affinities of Arundelemys dardeni, but the detailed description and comparative statements offered in our work may help to refine its phylogenetic position within the group. Additionally, we provide a description and homology interpretation of cranial scutes for Arundelemys dardeni. Cranial scutes are common in turtles and symplesiomorphically present in Arundelemys dardeni. As such, the scute pattern, which cannot be evaluated for all paracryptodires due to extensive skull sculpturing in many species, may provide future phylogenetic information for global paracryptodiran relationships.

We thank Donald Brinkman for initially sharing the CT scan, and Matthew Colbert for clarifying some scanning parameters as well as initially conducting the CT scan on which this work is based. We than reviewers Adán Pérez-García and Gabriel Ferreira for their excellent comments that helped improving this paper, as well as the editor Mark Young. The work is supported by a grant from the Swiss National Science Foundation (SNF 200021_178780/1).

Institutional Abbreviation

DORCM Dorset County Museum, Dorchester, United Kingdom.

USNM United States National Museum, Smithsonian Institution National Museum of Natural History, Washington, D.C., USA.

Additional Information and Declarations

Competing Interests

Author Contributions

Data Availability

The authors declare there are no competing interests.

Serjoscha W. Evers conceived and designed the experiments, performed the experiments, analyzed the data, prepared figures and/or tables, authored or reviewed drafts of the paper, and approved the final draft.

Yann Rollot and Walter G. Joyce conceived and designed the experiments, analyzed the data, authored or reviewed drafts of the paper, and approved the final draft.

The following information was supplied regarding data availability:

The CT dataset used to create our 3D models is available at MorphoSource: Heather Jamniczky provided access to these dataoriginally appearing in Lipka et al.2006, with data upload to MorphoSource funded by DBI-1902242. The files were downloaded from www.MorphoSource.org, Duke University. Media ID 000114666, DOI10.17602/M2/M114666.

The 3D models generated for our study:

Endosseous Labyrinth, Media ID 000351409, DOI10.17602/M2/M351409;

Facial nerve, Media ID 000351405, DOI10.17602/M2/M351405;

Carotid arteries, Media ID 000351401, DOI10.17602/M2/M351401;

Abducens nerves, Media ID 000351397, DOI10.17602/M2/M351397;

Right Quadrate, Media ID 000351393, DOI10.17602/M2/M351393;

Right Pterygoid, Media ID 000351389, DOI10.17602/M2/M351389;

Right Prootic, Media ID 000351385, DOI10.17602/M2/M351385;

Right Premaxilla, Media ID 000351381, DOI10.17602/M2/M351381;

Right Prefrontal, Media ID 000351377, DOI10.17602/M2/M351377;

Right Postorbital, Media ID 000351373, DOI10.17602/M2/M351373;

Right Parietal, Media ID 000351369,DOI10.17602/M2/M351369;

Right Palatine, Media ID 000351365, DOI10.17602/M2/M351365;

Right Opisthotic, Media ID 000351361, DOI10.17602/M2/M351361;

Right Nasal, Media ID 000351357, DOI10.17602/M2/M351357;

Right Maxilla, Media ID 000351353, DOI10.17602/M2/M351353;

Right Jugal, Media ID 000351349, DOI10.17602/M2/M351349;

Right Frontal, Media ID 000351337, DOI10.17602/M2/M351337;

Right Exoccipital, Media ID 000351333, DOI10.17602/M2/M351333;

Right Epipterygoid, Media ID 000351329, DOI10.17602/M2/M351329;

Left Quadrate,Media ID 000351295, DOI10.17602/M2/M351295;

Left Pterygoid, Media ID 000351290, DOI10.17602/M2/M351290;

Left Prootic, Media ID 000351286, DOI10.17602/M2/M351286;

Left premaxilla, Media ID 000351270, DOI10.17602/M2/M351270;

Left Prefrontal, Media ID 000351266, DOI10.17602/M2/M351266;

Left Postorbital, Media ID 000351258, DOI10.17602/M2/M351258;

Left Parietal, Media ID 000351247, DOI10.17602/M2/M351247;

Left Opithotic, Media ID 000351243, DOI10.17602/M2/M351243;

Left Nasal, Media ID 000351239, DOI10.17602/M2/M351239;

Left Maxilla, Media ID 000351235, DOI10.17602/M2/M351235;

Left Jugal, Media ID 000351231, DOI10.17602/M2/M351231;

Left Frontal, Media ID 000351227, DOI10.17602/M2/M351227;

Left Exoccipital, Media ID 000351223, DOI10.17602/M2/M351223;

Left Epipterygoid, Media ID 000351219, DOI10.17602/M2/M351219;

Vomer, Media ID 000351215, DOI10.17602/M2/M351215;

Supraoccipital, Media ID 000351211, DOI10.17602/M2/M351211;

Parabasisphenoid, Media ID 000351207, DOI10.17602/M2/M351211;

Basioccipital, Media ID 000351203, DOI10.17602/M2/M351203.

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
