# Peer review of "New interpretation of the cranial osteology of the Early Cretaceous turtle Arundelemys dardeni (Paracryptodira) based on a CT-based re-evaluation of the holotype"

_PeerJ, doi:10.7717/peerj.11495_

## Round 0.1 · original submission · Minor Revisions

Dear authors,

I have accepted the reviewers' decision of 'minor revisions'.

I look forward to receiving your revised manuscript.

·

Basic reporting

This is a descriptive work in which new anatomical information about a relatively poorly known taxon is provided, based on the revision of its holotype. The descriptions are adequate.
I recommend the publication of this work with new data on Arundelemys. I have made some minor suggestions in the attached PDF.

Experimental design

The methodology used to carry out this study is adequate, and it has been explained in detail. I have made a comment regarding a suggestion about this in the PDF.

Validity of the findings

The results are adequate, being based on the new data obtained through the redescription of the Arundelemys holotype.

·

Basic reporting

The manuscript presents a redescription of Arundelemys dardeni based on µCT reconstructions, as part of an effort by this group of coauthors to improve our basic anatomical knowledge of paracryptodiran turtles. The structure and writing of the manuscript are flawless (only a few typos are identified in the annotated PDF), all figures are high quality and relevant to the discussed subjects and the raw data is supplied in an online repository.

Experimental design

Classic thorough anatomical description, with relevant comparisons to other taxa.

Validity of the findings

Some structures are reinterpreted, new traits are described (e.g., head scute system) and everything is well based on the digital reconstruction presented in the manuscript, and the reader can confirm all observations in the original tomographic images, which are provided in an online repository.

Additional comments

I congratulate the authors for this straightforward, yet comprehensive redescription of this important taxon (and for the general effort to gather more information on paracryptodires as well). I provided only a few corrections of typos in the annotated PDF and have no issue with the interpretations presented in the manuscript, except for the homology hypothesis of some of the head scutes.

The midline series seem to be non-problematic in your hypothesis, as it is in all of the previous publications developing on this system. The more lateral scutes are a bit messier (aside from F, I and J-series, which, again, are undoubtedly identified). You suggested Mongolochelys scute identification in Sterli & de la Fuente 2013 is mistaken regarding D and H, but I believe the identification misrepresented in that study is that of Meiolania, instead: scutes D and H are switched there. In Proganochelys and Mongolochelys D is anterior to H, overlaying the parietal and postorbital, and contacting medially G and H, anteriorly F, laterally E and posterolaterally C. The H, on the other hand, on both taxa is posterior to D and overlays the squamosal, not reaching G nor F anteriorly. If we change H for D in Meiolania's identification this is the same pattern exactly, but if we switch D and H in Mongolochelys, it should also be changed in Proganochelys too. Further, in all subsequent publications (Rabi et al. 2013, 2014; Li et al. 2019) D is always in contact with F and G and never reaches scute A. In all previous publications scute H contacts X and A medially (except in Li et al. 2019 in which it is not identified, but there seems to be a posterior bifurcation of the X-D sulcus that could indicate a tentative H scute, again, between D, X and A as well). For these reasons, I think your scute D is better identified as H.
In all other studies, the only scutes to reach the orbit are F and J, but in your hypothesis, you have two differences: ?H and E also reach the orbit. In previous studies, F scutes are the only to be (i) restricted to the postorbital and (ii) margin the orbit posterodorsally. Both criteria are met by your E scute, which I propose is more parsimoniously interpreted as another of the F series. That would leave a ?H scute between two F-scutes. The three criteria you used to identify it as H also apply to an F-scute (in Annemys levensis one F also contacts G medially; Rabi et al. 2014). That leaves C as another doubt. C is only identified in Sterli & de la Fuente (2013), and it reaches E anteroventrally, D anteriorly, H medially and B posteriorly. It never reaches the scutes X or F (which would happen if you agree with my alternative interpretation). If we identify it instead as D, that would fit perfectly the original system: it contacts F anteriorly, X medially and H posteriorly.
To sum up, the alternative identification I propose is: scutes H, E, C and D in your hypothesis become F', F'', D and H.
Needless to say, homology hypotheses are just hypotheses, and you can disagree with my alternative interpretation. I consider it more parsimonious, though, and think you should at least consider it.

The authors are free to contact me if something is not clear in my assessment.
All the best regards,
Gabriel Ferreira

---

## Round 0.2 · accepted · Accept

Dear authors,

I have accepted your revised manuscript based on your response to the reviewer's document.

You will be contacted in due course by the production staff to take you through the proof stages.

Thank you again for choosing PeerJ as your publication venue, and I hope you will use us again in the future.